# The Constipation-Relieving Property of d-Tagatose by Modulating the Composition of Gut Microbiota

**DOI:** 10.3390/ijms20225721

**Published:** 2019-11-14

**Authors:** Yu-Xuan Liang, Peng Wen, Yu Wang, Dong-Mei OuYang, Da Wang, Yu-Zhong Chen, Ya Song, Jie Deng, Yuan-Ming Sun, Hong Wang

**Affiliations:** 1Guangdong Provincial Key Laboratory of Food Quality and Safety/College of Food Science, South China Agricultural University, Guangzhou 510642, China; l402972616@163.com (Y.-X.L.); wenpeng100@163.com (P.W.); ouydm2019@163.com (D.-M.O.); wangda134@foxmail.com (D.W.); songya_1990@163.com (Y.S.); scaudengjie@163.com (J.D.); 2Guangzhou Institute for Food Inspection, Guangzhou 511400, China; xxwangyu@163.com (Y.W.); Kittychenyuzhong@gmail.com (Y.-Z.C.)

**Keywords:** d-tagatose, prebiotic, constipation, gut microbiota, 16S rRNA, neurotransmitter

## Abstract

d-tagatose, a monosaccharide as well as a dietary supplement, has been reported as having a wide range of applicability in the food industry, however, the prebiotic activity, anticonstipation effects, and related mechanisms are still unclear. In this study, using the loperamide-induced constipation Kunming mice as the animal model, the effects of d-tagatose for the prevention of constipation were evaluated by gastrointestinal transit experiment and defecation experiment. Furthermore, the underlying mechanism was clarified by evaluating the change of the biochemical indicators and analyzing 16S rRNA amplicon of gut microbiota among the different mice groups. The results showed that the gastrointestinal transit rate, fecal number, and weight in six hours were significantly enhanced after the administration of d-tagatose. In addition, d-tagatose significantly increased the serum levels of acetylcholine (Ach) and substance P (SP), whereas the serum levels of nitric oxide (NO) were significantly decreased. Moreover, the 16S rRNA sequencing analysis revealed that the changes in the gut microbiota caused by constipation were restored by d-tagatose treatment. In conclusion, this study indicated that the administration of d-tagatose as a dietary supplement can effectively prevent and relieve constipation in Kunming mice, and it is a promising prebiotic candidate with constipation-relieving properties.

## 1. Introduction

Constipation is one of the most common health issues worldwide [1], and it is characterized by several symptoms such as infrequent bowel movements, small, dry and hard feces, and prolonged gastrointestinal transit time [2]. The cause of constipation is considered multifactorial [3]. Previous studies have shown that constipation is associated with imbalances in the composition of the gut microbiota [4,5]. For example, some common probiotics (*Lactobacilli* and *Bifidobacteria*) were generally reduced in constipation, while pathogenic bacteria (*Methanogenic archaea* and *Clostridia*) were increased in constipation [1,6,7]. Additionally, the enteric nervous system (ENS) plays a key role in the regulation of gastrointestinal motility, and the excitability of which is regulated by the levels of neurotransmitters [8]. Currently, the common way to relieve constipation is by using osmotic, secretagogues, and prokinetic drugs [9,10,11], however, the administration of these drugs is often accompanied by some side effects, such as irritancy and dependence [12]. Hence, an effective and easy-to-implement strategy to relieve constipation is desirable.

Recently, dietary supplements of natural products are regarded as a promising alternative to relieve constipation because of their safety and effectiveness [5]. For instance, some flavonoids are able to change the structure of epithelial cells and smooth muscle cells, or effectively regulate the quantitative alternations of the interstitial cells of Cajal and aquaporin 3 to relieve constipation in mice [13,14]. In addition, different doses of fructo-oligosaccharide, galacto-oligosaccharide, and isomalto-oligosaccharide have been proven to have the potential to regulate intestinal microbiota to relieve constipation [15]. Nevertheless, there are few studies on the ease of constipation by monosaccharides.

Tagatose, a kind of monosaccharide with a molecular weight of 180.16 Da, is recognized as a safe food ingredient by the China Food and Drug Administration (CFDA) and USA Food and Drug Administration (FDA) [16,17]. It has been reported that d-tagatose possesses low small intestinal digestibility (15~20%) [18,19], however, it can be utilized by the intestinal microflora to increase the concentration of short-chain fatty acids (SCFAs), lower the intestinal pH, and make the intestinal environment more harmonious [15,19,20]. A previous study found that the abundance of *Bacteroides*, *Lactobacillus*, and *Akkermansia* was significantly associated with the tagatose treatment in the colitis murine model [21]. In view of the aforementioned cause of constipation, we hypothesized that these characteristics of d-tagatose may confer beneficial effects on relieving constipation.

The aim of this study were as follows: (1) to evaluate the preventive effects of d-tagatose on the gastrointestinal transit and defecation status in mice and (2) to clarify d-tagatose underlying mechanisms of anticonstipation by analyzing the composition of gut microbiota and the indicators of serum, including acetylcholine (Ach), nitric oxide (NO), substance P (SP), and vasoactive intestinal peptide (VIP), in slow transit constipation mice.

## 2. Results

### 2.1. Effects of d-Tagatose on the Gastrointestinal Transit Rate

The effect of d-tagatose with different doses on gastrointestinal transit rates of mice is shown in Figure 1. Constipation symptoms were effectively induced by loperamide, indicating the constipation model was established successfully. The lowest gastrointestinal transit rate was in the model group, and the highest was in the high doses d-tagatose group. Although there was no significant difference between the low dose d-tagatose group and the model group, the medium and high dose d-tagatose groups showed significant acceleration on the gastrointestinal transit (*p* < 0.01). In particular, there was no significant difference in the gastrointestinal transit rate between the high-dose tagatose group (83.73%) and the blank group (83.42%), indicating that high-dose tagatose could restore gastrointestinal peristalsis to normal levels. These results indicate that the administration of d-tagatose could improve the intestinal peristalsis of constipated mice in a dose-dependent manner.

### 2.2. Effects of d-Tagatose on the Defecation Status

The effect of different doses of d-tagatose on the defecation status of mice is shown in Figure 2. In comparison to the model group, low and medium doses of d-tagatose treatment led to a significant increase in the fecal number and weight in six hours (*p* < 0.05), while in the high doses of d-tagatose group, the significant difference was observed only in the fecal weight in six hours. Regardless of the time under the action of low-, medium- and high-tagatose groups did not reach the level of the blank group, however, a decreased shorten time of the first black stool defecation (shortening rate to 28.53%, 31.09%, and 26.41%, respectively) than that of the model group was achieved. Overall, the administration of d-tagatose could shorten the average time of defecation; nevertheless, there were no significant differences between all d-tagatose groups and model group in the time to the first blank stool defecation.

### 2.3. Effects of d-Tagatose on Serum Neurotransmitter

The effects of d-tagatose on relieving constipation were further evaluated by measuring serum parameters in mice, including excitability neurotransmitters (Ach and SP), and the inhibitory neurotransmitters (NO and VIP). As shown in Table 1, the levels of Ach and SP were significantly increased in the d-tagatose treatment group (*p* < 0.05), while the levels of NO exhibited a significant decrease (*p* < 0.05). No statistical differences were observed in the levels of VIP between the d-tagatose group and model group. These results show that d-tagatose treatment effectively increased the levels of the excitability neurotransmitters (Ach and SP) and decreased the levels of the inhibitory neurotransmitters (NO) to speed up the motility of the gastrointestinal tract.

### 2.4. Effect of d-Tagatose on the Composition of Rectum Microbiota

To assess the influence of d-tagatose in the gut microbiome, we collected rectal content of three groups of mice, and for each group of 10 mice took six samples (a total of 18 rectum samples) and a community structure analysis was performed. All of the sequences were clustered with representative sequences, and a 97% sequence identity was obtained. Based on alpha and beta diversity measurement, overall association tests were then conducted.

From Figure 3, regarding the indexes of alpha diversity (the species richness, Chao-1 and Shannon, and the observed species), significant differences were observed between the blank and model group (*p* < 0.05). After the administration of d-tagatose, the above three indexes exhibited a significant increase in comparison with the model group (*p* < 0.05), while no significant differences were observed between the blank and d-tagatose group indicating the general diversity and richness of gut microbiota in constipated mice restored to normal. In addition, PCoA based on unweighted uniFrac matrixes revealed the beta diversity of the intestinal flora in mice. As shown in Figure 3, we can see that the data points shifted from the lower of the score plot to the upper in all of the d-tagatose treatment in comparison with the model group. Combined with the results of alpha diversity measurement, we can conclude that the administration of d-tagatose could restore the microbial richness and diversity in constipated mice.

### 2.5. The Relative Abundance of Rectum Microbiota

To further investigate the effect of tagatose treatment on the difference in the composition of the intestinal flora, the levels of phylum, family, and genus were further analyzed. All of the effective reads were clustered into 11 phyla, 30 families, and 35 genera using the Ribosomal Database Project (RDP) classifier. At the phylum level (Figure 4), the major dominant phyla in the rectum contents of mice were Bacteroidetes and Firmicutes, followed by Proteobacteria, Verrucomicrobia, Actinobacteria, Deferribacteres, Tenericutes, Saccharibacteria, Cyanobacteria and an unclassified group denoted as ‘‘other”. The core bacterium was regarded as the bacterium with the highest average relative abundance. At the phylum level, the core bacterium was Bacteroidetes, which were 77.97%, 82.90%, and 69.31% in the blank, model, and tagatose groups, respectively. In order to characterize the effect of tagatose on the difference of phylum level, the specific phylum levels with their relative abundance higher than 0.1% were selected and analyzed. As shown in Figure 4b, there were significant differences between Bacteroidetes and Proteobacteria. After treatment with loperamide, the mean abundance of Bacteroidetes in the model group was increased by 2.37% as compared with the blank group. However, under the action of tagatose, Bacteroidetes were then decreased by 14.01% as compared with the model group (*p* < 0.05), and there was no significant difference as compared with the blank group. By contrast, an increase in the level of Proteobacteria after the d-tagatose treatment was also observed as compared with the model group. These results indicated that d-tagatose could improve the change of gut microbiota in the levels of Bacteroidetes and Proteobacteria caused by loperamide.

At the family level, the major dominant families in the rectum contents of mice were BacteroidalesS24-7group (phylum Bacteroidetes), Prevotellaceae (phylum Bacteroidetes), Bacteroidaceae (phylum Bacteroidetes), and Lachnospiraceae (phylum Firmicutes), followed by twenty-six families, which are shown in Figure 5. It can be seen that the core bacterium was BacteroidalesS24-7group, and its abundance in the blank, model, and tagatose group were 45.81%, 49.68%, and 33.55%, respectively. Similarly, the specific family levels with their relative abundance higher than 0.1% were selected and analyzed, and results showed that six families’ levels revealed significant differences (Figure 6). In comparison to the blank group, the structure of intestinal flora was changed in the model group. For example, the mean abundance of BacteroidalesS24-7group and Lactobacillaceae increased in the model group, while the mean abundance of Lachnospiraceae, Porphyromonadaceae, Corynebacteriaceae, and Erysipelotrichaceae reduced in the model group. However, after administration of d-tagatose, the trends of six families had been changed. The relative abundance of Erysipelotrichaceae, Lachnospiraceae, Porphyromonadaceae, and Corynebacteriaceae was significantly increased as compared with the model group (*p* < 0.05), while the relative abundance of BacteroidalesS24-7group significantly reduced (*p* < 0.01). These results reveal that the administration of d-tagatose could effectively relieve the change of gut microbiota caused by loperamide in the above mentioned six families.

At the genus level (Figure 7), the major dominant genera in the rectum contents of mice were *Bacteroides* (phylum Bacteroidetes, family Bacteroidaceae) and *Alloprevotella* (phylum Bacteroidetes, family Prevotellaceae), followed by thirty-three genera. As shown in Figure 7, the core bacterium of the genus level was *Bacteroides* with the levels of 13.42%, 10.86%, and 18.12% in the blank, model, and tagatose groups, respectively. Similar to the analysis of phylum and family levels, we found that twelve genus levels exhibited significant differences (Figure 8). Among them, after treatment with loperamide, the average abundance of *Bacteroides, Alistipes*, *Parabacteroides*, *Eubacterium xylanophilum group*, *Paenalcaligenes*, *Facklamia*, *Atopostipes*, *Clostridium innocuum group*, and *RuminococcaceaeNK4A-214group* were reduced, while the average abundance of *Lactobacillus* was increased. Interestingly, the trends of ten genera had been reversed with the administration of d-tagatose. As presented in Figure 8, the relative abundance of *Bacteroides*, *Alistipes*, *Desulfovibrio*, *RuminococcaceaeTCG-004,* and *RuminococcaceaeNK4A-214group* was significantly increased in the d-tagatose group (*p* < 0.05), in particular, the relative abundance of *Parabacteroides*, *Paenalcaligenes*, *Facklamia*, *Atopostipes,* and *Clostridium innocuum group* highly significantly increased in the d-tagatose group (*p* < 0.01). These results indicate the beneficial effect of d-tagatose on the levels of ten genera, which were influenced by loperamide.

### 2.6. Correlation between Intestinal Microflora and Biological Indexes

The correlation of the top 40 genera and the biological indexes were performed by Spearman correlation analysis (Figure 9). Among them, 13 genera significantly exhibited a correlation with the biological indexes of constipation. As the indicators of constipation, for instance, the gastrointestinal transit rate was positively correlated with *Bacteroides* and *Enterobacter* (*p* < 0.05). Conversely, it was negatively correlated with *FamilyXIII AD3011 group* and *Eubacterium coprostanoligenes group* (*p* < 0.05). The fecal number in six hours and fecal weight in six hours were negatively correlated with *RuminococcaceaeTCG−004*, *Lactobacillus*, and *Eubacterium coprostanoligenes group* (*p* < 0.05), while they were positively related to *Enterobacter* (*p* < 0.01). In addition, the time to the first black stool defecation was negatively correlated with *Enterobacter* (*p* < 0.05). For neurotransmitters, there was a significant negative correlation between *Enterobacter* and VIP (*p* < 0.05), and *Helicobacter*, *Corynebacterium1*, *Desulfovibrio* were also negatively with the NO (*p* < 0.05). *Akkermansia* and *Eubacterium coprostanoligenes group* were negatively correlated with the SP (*p* < 0.05), while *Helicobacter* was positively correlated with the SP (*p* < 0.05). These results indicate that biological indicators of constipation, such as intestinal movement, defecation, and level of the related neurotransmitter, are significantly correlated with specific bacteria. Thus, changes in the abundance of specific bacteria may be beneficial to alleviating or aggravating constipation.

## 3. Discussion

Many studies have shown that the indigestible carbohydrates of a supplement possess beneficial effects on gut health [22,23,24]. These carbohydrate supplements are mainly polysaccharides and oligosaccharides, while there are few studies about monosaccharides. d-tagatose is a monosaccharide with 92% sweetness of sucrose but only 38% of the calorie, the primary features of which include a lower absorption rate in the small intestine but utilization by the intestinal flora [18,19,25]. Importantly, d-tagatose has not been studied in relieving constipation. Hence, based on the indigestible properties of tagatose and the activity of other similar functional carbohydrates in the management of relieving constipation, the effect of d-tagatose on the constipation was investigated using a constipation animal model.

To establish the model, loperamide was used to induce constipation, the primary mechanism of which can be attributed to the inhibition of the release of Ach and prostaglandins, thus, leading to the reduced frequency of bowel motility and prolonged bowel movement [1]. The gastrointestinal transit, time to the first black stool defecation, as well as fecal weight and number in unit time are key indices with which to evaluate the function of the gastrointestinal transit [1,26]. The movement of gastrointestinal contents is mainly promoted by peristalsis, which can be detected by measuring the gastrointestinal transit rate and time to the first black stool defecation [3]. In addition, the fecal weight and number in six hours are also considered as direct indicators for reflecting the status of constipation. This study revealed that d-tagatose administration effectively accelerated gastrointestinal peristalsis to prevent constipation in a dose-dependent manner (Figure 1). This may be related to the lower intestinal absorption of tagatose, allowing it to pass through the small intestine faster, as well as, in the defecation experiment, the administration of tagatose could enhance the fecal number and weight in six hours (*p* < 0.05). In addition, it can be seen that both the GI transit rate and the time to the first stool defecation were improved after d-tagatose treatment. For the GI transit rate, it was shown to be restored upon tagatose treatment. However, for the time to first stool, although a higher level still existed as compared with the blank group, there was a reduction of 28.53%, 31.09%, and 26.41% by the treatment of the low-, medium- and high-tagatose, respectively as compared with the model group (Figure 2c). Different measurement methods and doses of loperamide may be responsible for the different results of the gastrointestinal transit and the time to the first black stool defecation. In a word, these results may be explained by the restoring of gut diversity, and thus the improvement of gut microbiota to promote the intestinal peristalsis. Combined with the Spearman correlation analysis (Figure 9), the parameters of relieving constipation were significantly correlated with the specific bacterium (such as *Bacteroides*). This was consistent with the previous reports that the indexes of constipation were significantly associated with the level of the specific bacterium [3,27].

Neurotransmitter also plays an important role in gastrointestinal hormones [28]. A couple of neurotransmitters have been identified to be associated with the motor activity of the gastrointestinal tract, such as Ach, NO, SP, and VIP, which play important roles in the regulation of gastrointestinal motility [14,24,29]. Among them, Ach and SP are excitatory neurotransmitters that promote bowel motility and speed defecation, while NO and VIP are inhibitory neurotransmitters that relax the gastrointestinal tract and inhibit bowel movement [3,30]. Disruption of the balance between these excitatory and inhibitory neurotransmitters may lead to constipation. Our study revealed that the administration of d-tagatose effectively regulated the excitability of the ENS, as the levels of excitatory neurotransmitters, Ach and SP, were increased, while the level of inhibitory neurotransmitter NO was decreased (Table 1). These results are in line with the previous report, suggesting that the regulation in neurotransmitters plays a crucial role in intestinal peristalsis [3].

The 16S rRNA amplicon of gut microbiota was applied to investigate the underlying mechanism among the different mice groups. First, the overall association tests by the alpha and beta diversity measurements were conducted to provide a holistic view of the microbiota structure [26]. From Figure 2, we can see that the administration of d-tagatose changed alpha diversity based on species richness (Chao-1, observed species, and Shannon), and a significant change in the structure of the rectum microbiota was observed in a PCoA plot of beta diversity (Figure 2). Moreover, there were significant differences among the blank, model, and d-tagatose groups, suggesting that the diversity of general intestinal flora induced by loperamide-constipation was restored after d-tagatose treatment.

Bacteroidetes were the dominant phylum in our study. In this study, as compared with the model group, a significant decrease in the abundance of Bacteroidetes was exerted in the d-tagatose group (Figure 3, *p* < 0.01). Since a previous study reported that a lower abundance of Bacteroidetes was observed in healthy than constipated patients [31], and the intake of dietary fiber was beneficial to the decrease of Bacteroidetes [32], in this study, the results indicated that d-tagatose has a beneficial function as a dietary fiber to improve gut microbiota.

At the family level (Figure 4), this study revealed that the relative abundance of Lachnospiraceae, Porphyromonadaceae, and Erysipelotrichaceae were significantly increased after administration of d-tagatose (*p* < 0.05). It was reported that the increases of Lachnospiraceae, Porphyromonadaceae, and Erysipelotrichaceae were beneficial for the production of SCFAs, which play an important role in protecting the host from infection and maintaining the normal physiological function of the gut [33,34]. Furthermore, studies have shown that SCFAs can also decrease intestinal pH and promote colon motility [35,36]. For instance, Lachnospiraceae is a family of clostridia [37,38], which is associated with the production of butyric acid [39,40]. Erysipelotrichaceae is another important butyrate-producing member [41], and Porphyromonadaceae can alleviate colonic inflammation [42,43]. Therefore, we hypothesized that d-tagatose treatment could promote the proliferation of SCFAs-producing bacteria, thus accelerating the intestinal peristalsis. In addition, the Lactobacillaceae and Bifidobacteriaceaec contain some beneficial common genera of bacteria, such as *Lactobacillus* and *Bifidobacterium*; in this study, higher abundance of Lactobacillaceae was found in constipated groups than other groups, while Bifidobacteriaceae was not detected. Ren et al. reported that the polysaccharides of *Enteromorpha* could relieve loperamide-induced constipation; similarly, the highest level of Lactobacillaceae was found in constipated groups and *Bifidobacterium* was not detected in the treatment group [44]. The characteristics of constipation-relieving properties may not necessarily be shown in the increased level of Lactobacillaceae and Bifidobacteriaceae, and thus further study needs to investigate the cause of such a phenomenon.

At the genus level, previous studies reported that *Bacteroides* is the major portion of the mammalian gut microbiota, which plays an important effect on the processing of complex molecules to simpler ones [45,46,47]. As documented in the literature, the species of *Bacteroides* in the gut were devoted to the uptake and breakdown of indigestible carbohydrates [47,48,49]. As expected, *Bacteroides* was the dominant genus in the gut microbiota of mice, and its content was significantly increased after administration of d-tagatose (Figure 7, *p* < 0.01), suggesting that the higher levels of *Bacteroides* could be beneficial to produce more metabolite (SCFAs), as well as accelerate intestinal peristalsis [35,36]. In addition, the relative abundances of *Parabacteroide*, *RuminococcaceaeTCG-004,* and *RuminococcaceaeNK4A-214group* increased after administration of d-tagatose (Figure 7, *p* < 0.05), which were positively associated with the production of SCFAs [50,51,52]. In a word, it can be concluded that for the intestinal flora diversity was restored after tagatose administration. Regarding the composition, changes in core bacteria (reduction of Bacteroidetes and an increase in *Bacteroides*) were observed, which are conducive to forming a harmonious intestinal environment and promoting the relief of constipation.

## 4. Materials and Methods

### 4.1. Chemicals and Reagents

The d-tagatose was purchased from Jcantek Pharmaceuticals Ltd. (Wuxi, China), and the content of the d-tagatose was 98%. Loperamide hydrochloride (Xi’an Janssen Pharmaceutical Ltd., Xi’an, China) was purchased from Dashenlin drugstore. Enzyme-linked immunosorbent assay (ELISA) kits were used to measure the levels of Ach, SP and VIP were purchased from Jianglai industrial Ltd. (Shanghai, China). The kit for the determination of the level of NO was purchased from Nanjing Jiancheng Bioengineering Institute (Nanjing, China). The Microbial DNA was extracted from rectum contents samples using the E.Z.N.A.^®^ Soil DNA Kit (Omega Bio-tek, Norcross, GA, USA). Amplicons were extracted and purified using the AxyPrep DNA Gel Extraction Kit (Axygen Biosciences, Union City, CA, USA). The methods were adopted to prepare loperamide hydrochloride and activated carbon meal solution [1].

### 4.2. Animals and Experimental Design

Specific Pathogen Free (SPF) male Kunming mice (initial weight 18~22 g) were purchased from Guangdong Medical Laboratory Animal Centre (Guangzhou, China); the mice were fed under standard conditions at a room temperature of 25 ± 2 °C and humidity of 50% ± 5% with a 12 h light-dark cycle. They were fed with standard commercial mouse food (containing 64% carbohydrate, 19% protein, and 17% fat), and water was provided ad libitum. All of the protocols were approved by the Ethics Committee of South China Agricultural University (SYXK2014-0136), and the approval number was 2017009.

#### 4.2.1. The Gastrointestinal Transit Experiment

The methods were conducted by the Technical Standards for Testing and Assessment of Health Food formulated by the Chinese Ministry of Health. After 7 days of adaptive feeding, the 50 mice were randomly divided into five groups, each group consisting of ten animals. All animals were administered orally by gavage. A dose of 5 g/day is the recommended dietary allowance of d-tagatose for a human being weighing 60 kg, which is equivalent to 0.083 g/kg per day. The dose groups set for the mice were 5, 10, and 20 times the equivalent recommended dietary allowance of prebiotics for humans. The blank group and model group were given distilled water via gavage once per day for 7 consecutive days; the low, medium, and high dose d-tagatose groups received 0.6 mL of 0.43 g/kg BW, 0.85 g/kg BW, or 1.70 g/kg BW d-tagatose solutions, respectively, in the same manner as the blank and model groups for 7 days. Then, all groups fasted overnight for 16 h (water was not restricted) before measurement of the indicator. The model and d-tagatose groups were treated with loperamide (5 mg/kg BW, 0.6 mL) via gavage to induce constipation, and the blank group was treated with distilled water (0.6 mL). Thirty minutes later, all groups were treated with activated carbon meal solution containing the corresponding content. At the end of the experiment, the gastrointestinal transit rate (the rate of carbon powder propulsion through the small intestine) was measured by the methods in [53,54].

#### 4.2.2. The Experiment of Defecation

The methods were conducted by the Technical Standards for Testing and Assessment of Health Food formulated by the Chinese Ministry of Health. The process for defecation experiment was the same with the gastrointestinal transit experiment, while the model and d-tagatose groups were treated with loperamide (10 mg/kg BW, 0.6 mL) according to the dose of Ren [44]. At the end of the experiment, all of the mice were used to examine defecation status including the time of first black stool defecation, fecal numbers in six hours, and fecal weight in six hours [55].

#### 4.2.3. The Slow Transit Constipation Experiment

The experiment was conducted according to the method of Zhu with slight modifications [56]. After 7 days of adaptive feeding, the mice were randomly divided with ten mice into each of the three groups. All the animals were administered orally by gavage. Both the model and d-tagatose groups were treated with 10 mg/kg BW dose of loperamide for the first two weeks to induce the model of slow transit constipation mice, and the blank group was treated with distilled water. Then, for the following two weeks, the blank and model groups were treated with distilled water, and the d-tagatose group received 1.70 g/kg d-tagatose solutions, respectively. At the end of the experiment, the mice were killed with light ether anesthesia. Blood samples were collected and centrifuged at 3000 r/min at 4 °C for 15 min to obtain serum. The rectum contents of each mouse were collected and stored at −80 °C until analysis.

### 4.3. Determination of Ach, NO, SP, and VIP Levels in Serum

The levels of Ach, SP, and VIP in the serum were measured using ELISA kits, and the levels of NO in the serum were determined by commercially available diagnostic kits. All measurement steps were carried out according to the manufactuerers’ instructions.

### 4.4. DNA Extraction, PCR, and 16S rDNA Sequencing

The microbial DNA was extracted from rectum contents samples using the E.Z.N.A.^®^ Soil DNA Kit by the protocols. The V4-V5 region of the bacterial 16S ribosomal RNA gene was amplified by PCR using primers 515F 5′-GTGCCAGCMGCCGCGG-3′ and 907R 5′-CCGTCAATTCMTTTRAGTTT-3′, where the barcode is an eight-base sequence unique to each sample. The PCR reactions were performed in triplicate 20 μL mixtures containing 4 μL of 5× FastPfu Buffer, 2 μL of 2.5 mM dNTPs, 0.8 μL of each primer (5 μM), 0.4 μL of FastPfu Polymerase, and 10 ng of template DNA.

Amplicons were extracted from 2% agarose gels and purified using a AxyPrep DNA Gel Extraction Kit according to the instructions. Purified amplicons were pooled in equimolar, and sequenced using paired-end sequencing (2 × 250) on the HiSeq 2500 platform according to the standard protocols. The raw reads were deposited into the NCBI Sequence Read Archive (SRA) database (accession number: SRP041836).

### 4.5. Analysis of Sequences and Relative Abundance of Microbiota Community Members

The raw sequences were trimmed and filtered by using the QIIME software (Version1.9). The high-quality reads were merged to generate the 16S rDNA V4 fragment sequences using FLASH software. Chimera sequences were filtered out using the Gold database by UCHIME (Version 4.2.40). All quality-filtered sequences were then clustered into operational taxonomic units (OTUs) with a threshold of 97% sequence similarity, by utilizing UPARSE software (Version 7.0). The representative sequences for each OTU were taxonomically assigned to the Silva database (16S rDNA) and unite database (ITS) using the RDP classifier [57]. Then, OTUs were processed by removing chloroplast sequences, chondriosome sequences, and unclassified sequences. The normalized OTU abundance profile was generated by utilizing a standard sequence number corresponding to the sample with the least sequences.

On the basis of the normalized OTU abundance profile, the three alpha diversity indices (Chao1, observed species, and Shannon) were calculated to estimate the species diversity and richness of each sample [58]. Subsequently, the differences of samples in OTU-level were evaluated through the principal co-ordinates analysis (PCoA) based on Bray-Curtis by using R software [59].

### 4.6. Statistical Analysis

All statistics were analyzed by using GraphPad Prism 7. The data are presented as mean ± SD for each group. The differences between the mean values of the groups were analyzed by one-way analysis of variance using (least significant difference, LSD) multiple range tests. If the variance was not uniform, and the non-parametric test was used (Kruskal-Wallis). The analyses were performed using SPSS software (version 20.0; SPSS Inc., Chicago, IL, USA). A *p*-value of less than 0.05 was considered to indicate statistical significance.

## 5. Conclusions

In summary, the results showed that the key constipation indicators (the gastrointestinal transit rate, fecal number in six hours, and fecal weight in six hours) were significantly enhanced after the administration of d-tagatose, suggesting that different doses of d-tagatose can prevent constipation symptoms in mice. In addition, the administration of d-tagatose treatment was beneficial for restoring the diversity of gut microbiota, improving the composition of gut microbiota (mainly for phylum of Bacteroidetes), and influencing the ENS by regulating the balance between the inhibitory and excitatory neurotransmitters (Ach, SP, and NO). Therefore, this study indicates that the administration of d-tagatose as a dietary supplement could effectively relieve constipation in mice, and it is a promising prebiotic candidate for the food industry. Further studies are required to study the process change of defecation status and gut microbiota during the intervention and the relationship between the SCFAs and intestinal peristalsis.

## Figures and Tables

**Figure 1 ijms-20-05721-f001:**
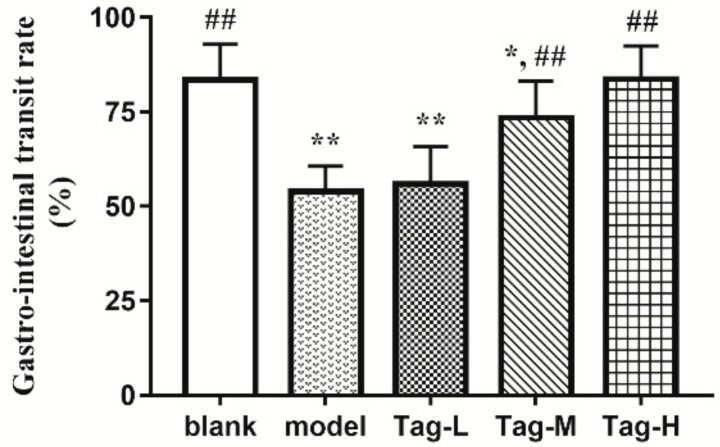
Effects of d-tagatose on the gastrointestinal transit rate of mice. Blank group (days 1–7, distilled water administration period and day 8, distilled water administration but no induction of constipation); model group (days 1–7, distilled water administration period and day 8, distilled water administration and induction of constipation); tagatose group (days 1–7, tagatose administration period and day 8, tagatose administration and induction of constipation, Tag-L mice treated with 0.43 g/kg body weight (BW) tagatose, Tag-M mice treated with 0.85 g/kg BW tagatose, and Tag-H mice treated with 1.70 g/kg BW tagatose). *, compared with the blank group, *p* < 0.05; **, compared with the blank group, *p* < 0.01; and ^##^, compared with the model group, *p* < 0.01.

**Figure 2 ijms-20-05721-f002:**
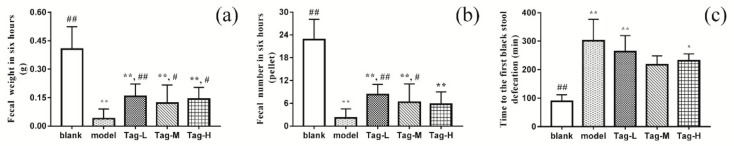
Effects of d-tagatose on the defecation status of mice. (**a**) Fecal weight in six hours, (**b**) fecal number in six hours, and (**c**) time to the first black stool defecation. Blank group (days 1–7, distilled water administration period and day 8, distilled water administration but no induction of constipation); model group (days 1–7, distilled water administration period and day 8, distilled water administration and induction of constipation); and tagatose group (days 1–7, tagatose administration period and day 8, tagatose administration and induction of constipation, Tag-L mice treated with 0.43 g/kg BW tagatose, Tag-M mice treated with 0.85 g/kg BW tagatose, and Tag-H mice treated with 1.70 g/kg BW tagatose). *, compared with the blank group, *p* < 0.05; **, compared with the blank group, *p* < 0.01; ^#^, compared with the model group, *p* < 0.05; ^##^, compared with the model group, *p* < 0.01.

**Figure 3 ijms-20-05721-f003:**
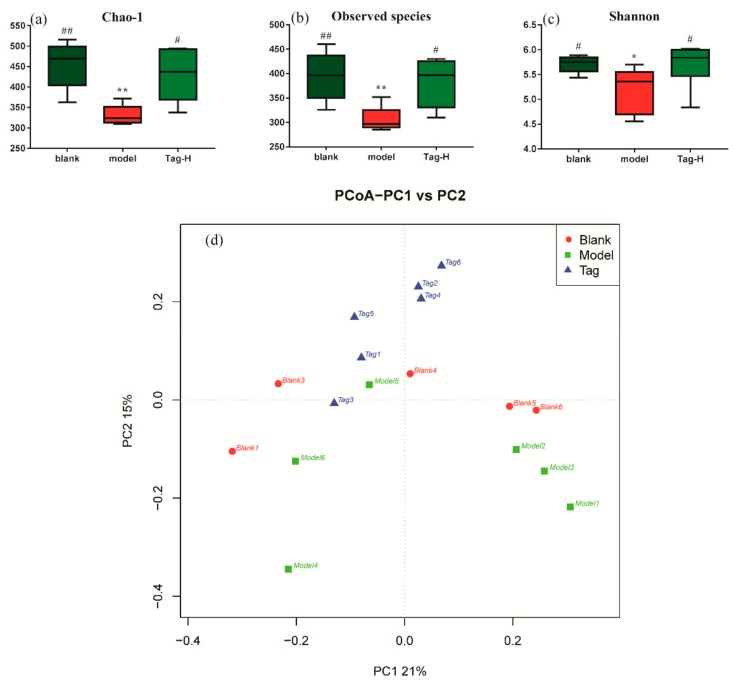
Alpha diversity indexes and beta diversity indexes in mice. (**a**) Chao-1 (alpha diversity index), (**b**) observed species (alpha diversity index), (**c**) Shannon (alpha diversity index), and (**d**) PCoA plots based on unweighted uniFrac metrics (beta diversity index). Blank group (days 1–14, distilled water administration period and days 15–28, distilled water administration period); model group (days 1–14, induction of constipation period and days 15–28, distilled water administration period); tagatose group (days 1–14, induction of constipation period and days 15–28, tagatose administration period, Tag-H mice treated with 1.70 g/kg BW tagatose). *, compared with the blank group, *p* < 0.05; **, compared with the blank group, *p* < 0.01; ^#^, compared with the model group, *p* < 0.05; and ^##^, compared with the model group, *p* < 0.01.

**Figure 4 ijms-20-05721-f004:**
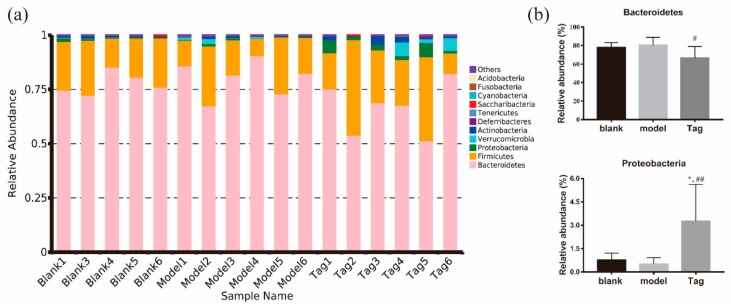
Community abundance on phyla level. (**a**) Changes in the level of phyla in different groups and (**b**) changes of the abundance of selected phyla in different groups (relative abundance more than >0.1%). Blank group (days 1–14, distilled water administration period and days 15–28, distilled water administration period); model group (days 1–14, induction of constipation period and days 15–28, distilled water administration period); and tagatose group (days 1–14, induction of constipation period and days 15–28, tagatose administration period, Tag-H, mice treated with 1.7 g/kg BW tagatose). *, compared with the blank group, *p* < 0.05; ^#^, compared with the model group, *p* < 0.05; and ^##^, compared with the model group, *p* < 0.01.

**Figure 5 ijms-20-05721-f005:**
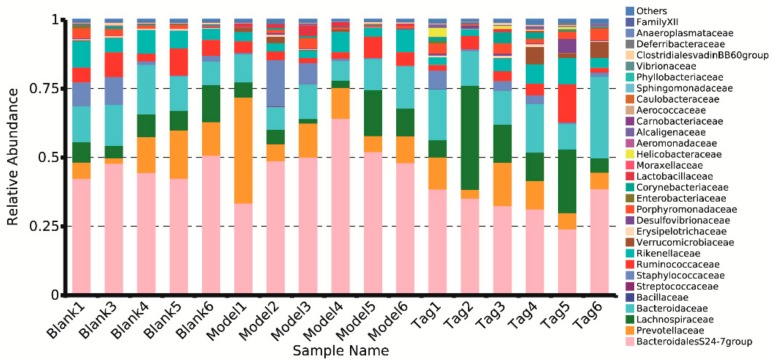
Changes in the level of families in different groups. Blank group (days 1–14, distilled water administration period and days 15–28, distilled water administration period); model group (days 1–14, induction of constipation period and days 15–28, distilled water administration period); and tagatose group (days 1–14, induction of constipation period and days 15–28, tagatose administration period, Tag-H, mice treated with 1.70 g/kg BW tagatose).

**Figure 6 ijms-20-05721-f006:**
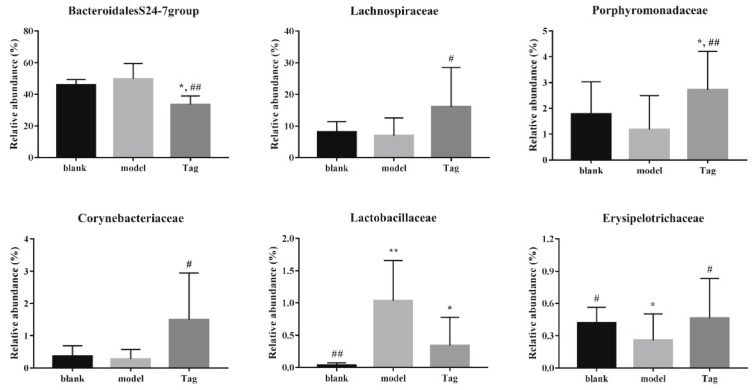
Changes of the abundance of selected families in different groups (relative abundance more than >0.1%). Blank group (days 1–14, distilled water administration period and days 15–28, distilled water administration period); model group (days 1–14, induction of constipation period and days 15–28, distilled water administration period); and tagatose group (days 1–14, induction of constipation period and days 15–28, tagatose administration period, Tag-H mice treated with 1.70 g/kg BW tagatose). *, compared with the blank group, *p* < 0.05; **, compared with the blank group, *p* < 0.01; ^#^, compared with the model group, *p* < 0.05; and ^##^, compared with the model group, *p* < 0.01.

**Figure 7 ijms-20-05721-f007:**
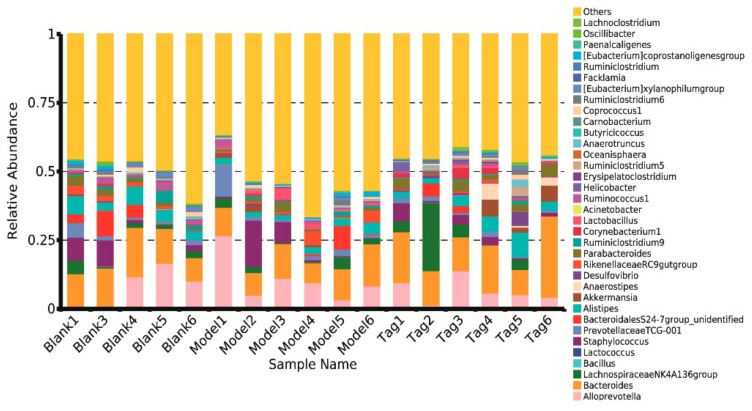
Changes in the level of genera in different groups. Blank group (days 1–14, distilled water administration period and days 15–28, distilled water administration period); model group (days 1–14, induction of constipation period and days 15–28, distilled water administration period); and tagatose group (days 1–14, induction of constipation period and days 15–28, tagatose administration period, Tag-H mice treated with 1.70 g/kg BW tagatose).

**Figure 8 ijms-20-05721-f008:**
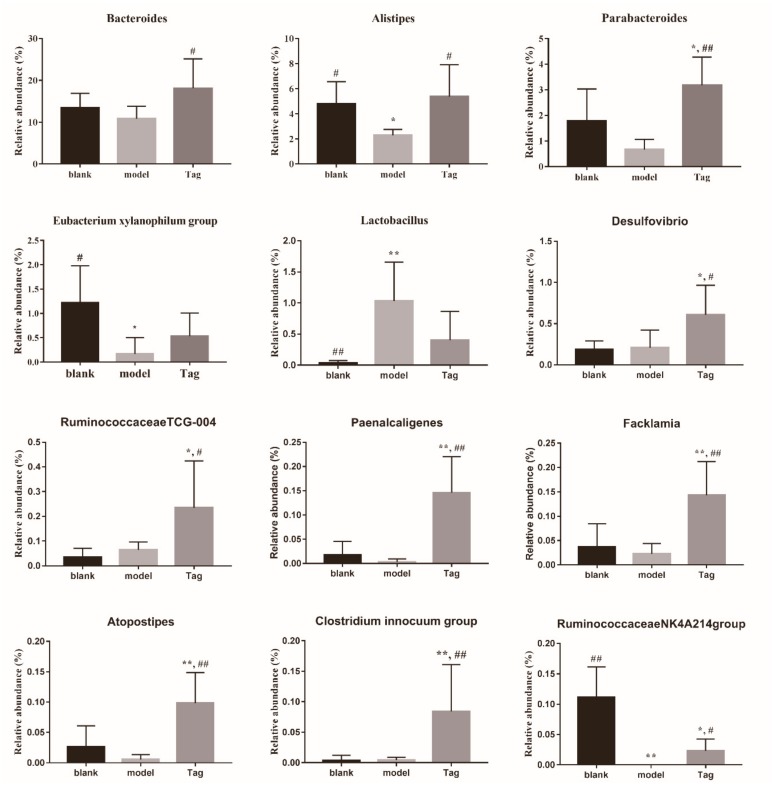
Changes of the abundance of selected genera in different groups (relative abundance more than > 0.1%). Blank group (days 1–14, distilled water administration period and days 15–28, distilled water administration period); model group (days 1–14, induction of constipation period and days 15–28, distilled water administration period); and tagatose group (days 1–14, induction of constipation period and days 15–28, tagatose administration period, Tag-H mice treated with 1.70 g/kg BW tagatose). *, compared with the blank group, *p* < 0.05; **, compared with the blank group, *p* < 0.01; ^#^, compared with the model group, *p* < 0.05; ^##^, compared with the model group, *p* < 0.01.

**Figure 9 ijms-20-05721-f009:**
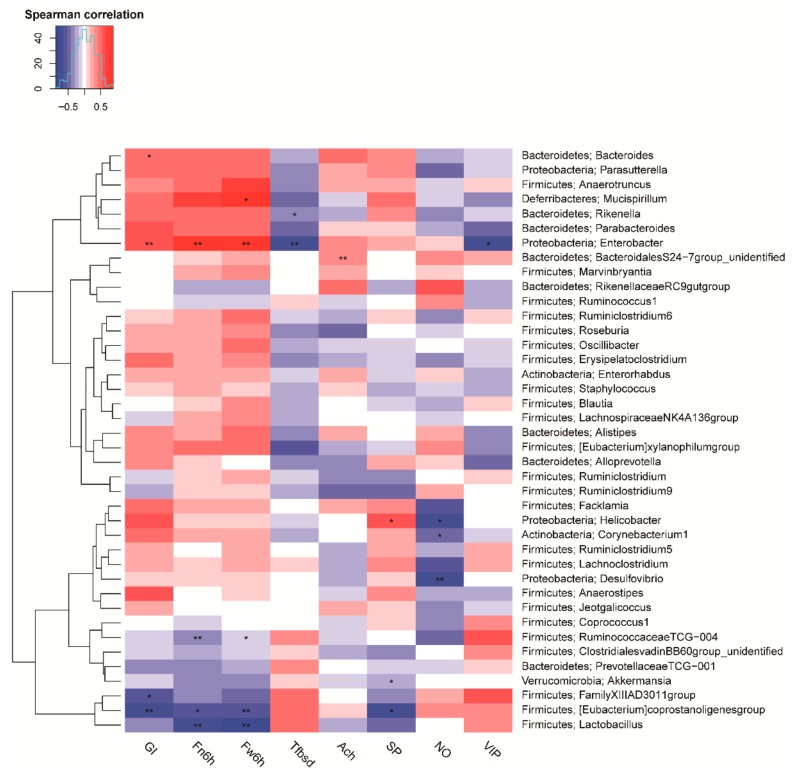
Spearman correlation analysis of top 40 genera and biological indicators. *, *p* < 0.05, **, *p* < 0.01. GI, gastrointestinal transit rate; Fn6h, fecal number in six hours; Fw6h, fecal weight in six hours; Tfbsd, time to the first black stool defecation; Ach, acetylcholine; SP, substance P; NO, nitric oxide; and VIP, vasoactive intestinal peptide.

**Table 1 ijms-20-05721-t001:** Effects of d-tagatose on serum parameters in mice.

Group	Ach (pg/L)	NO (μmol/L)	VIP (pg/L)	SP (pg/L)
blank	38.17 ± 10.17 ^#^	30.71 ± 3.81	18.19 ± 3.75 ^##^	31.97 ± 12.81
model	27.82 ± 7.28 *	26.99 ± 4.77	26.19 ± 4.53 **	24.76 ± 6.23
Tag-H	37.98 ± 11.58 ^#^	15.60 ± 2.76 *^,##^	24.07 ± 2.50	49.11 ± 11.25 ^##^

Blank group (days 1–14, distilled water administration period and days 15–28, distilled water administration period); model group (days 1–14, induction of constipation period and days 15–28, distilled water administration period); and tagatose group (days 1–14, induction of constipation period and days 15–28, tagatose administration period). Tag-H, mice treated with 1.70 g/kg BW tagatose). *, compared with the blank group, *p* < 0.05; **, compared with the blank group, *p* < 0.01; ^#^, compared with the model group, *p* < 0.05; and ^##^, compared with the model group, *p* < 0.01. Acetylcholine (Ach); nitric oxide (NO); substance P (SP); vasoactive intestinal peptide (VIP).

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
