# Peer review of "The Constipation-Relieving Property of d-Tagatose by Modulating the Composition of Gut Microbiota"

_ijms, 2019, doi:10.3390/ijms20225721_

Round 1

Reviewer 1 Report

In this paper the authors investigate the use of tagatose as a prebiotic to treat constipation in a murine model. The authors demonstrate that the use of tagatose improves bowel movement and increases microbiome diversity.

The introduction could be improved by expanding upon microbiota changes during constipation and more specific information on how tagatose is known to alter the microbiota. The role of Ach, NO, SP and VIP in bowel function should also briefly be mentioned. Further a brief description of how the model works, i.e. how loperamide induces constipation would be useful.

The results are generally well presented. However, in figure 1 the GI transit rate is shown to be restored upon tagatose treatment but in figure 2c the time to first stool is still increased in the tagatose treated groups. Please explain how transit time can be restored but time to first stool not be?

In line 252, the microbiota is described as being restored after tagatose treatment. Whilst the data does show increased alpha diversity in the treated group. The composition of microbiota is different from the untreated control group. Suggesting bacterial diversity has been restored but the composition of bacterial species has been altered. 

The discussion is quite descriptive of the results in nature and would benefit by placing the results in context with published literature more. Especially, what is known about the use of carbohydrate supplementation during constipation and how this study adds to this.

Author Response

Dear Editor/Reviewer:

We are very grateful for your helpful suggestions for revision (Manuscript ID: ijms-617693). The manuscript has been carefully revised according to the editor and reviewers’ comments. All the changes are marked in red fonts throughout the revised manuscript. The issues raised by the editor and reviewers have been addressed as follows. Page, line or figure numbers refer to the revised manuscript. 

Point 1: The introduction could be improved by expanding upon microbiota changes during constipation and more specific information on how tagatose is known to alter the microbiota.

Response 1: According to the reviewer’s suggestion, the introduction could be improved and more information had been added in the revised manuscript (page 1, line 33-37; page 2, line 51-57).

Point 2: The role of Ach, NO, SP and VIP in bowel function should also briefly be mentioned.

Response 2: According to the reviewer’s suggestion, the corresponding information had been added in the revised manuscript (page 10, line 300-308).        

Point 3: Further a brief description of how the model works, i.e. how loperamide induces constipation would be useful.

Response 3: According to the reviewer’s suggestion, a brief description of the induced constipation by loperamide had been added in the revised manuscript (page 10, line 272-275).

Point 4: The results are generally well presented. However, in figure 1 the GI transit rate is shown to be restored upon tagatose treatment but in figure 2c the time to first stool is still increased in the tagatose treated groups. Please explain how transit time can be restored but time to first stool not be?

Response 4: Thank you for the review’s suggestion. We can see that both the GI transit rate and the time to the first stool defecation were improved after D-tagatose treatment. For the GI transit rate, it was shown to be restored upon tagatose treatment (Figure 1). However, for the time to first stool, although a higher level was still existed compared to the blank group, there was a reduction of 28.53%, 31.09%, and 26.41% by the treatment of the low-, medium- and high-tagatose respectively in comparison with the model group (Figure 2c). There were two possible reasons for this result. Firstly, the determination methods of the two experiments are different. For the GI transit rate experiment, it was calculated according to the rate of carbon powder propulsion through the small intestine, while the first black stool defecation was calculated based on the transit time passing through the entire gastrointestinal tract. Therefore, the results presented in the two experiments were somewhat different. Secondly, the dose of loperamide in the GI transit experiment was 5 mg/kg, while the dose of loperamide in the defecation experiment was 10 mg/kg, which was adopted according to the Technical Standards for Testing & Assessment of Health Food formulated by the Chinese Ministry of Health. Constipation caused by higher doses of loperamide is more severe and more difficult to recover. Hence, in figure 1 the GI transit rate is shown to be restored upon tagatose treatment, but in figure 2c the time to first stool is still increased in the tagatose treated groups. The corresponding information had been added in the revised manuscript (page 10, line 282-296).

Point 5: In line 252, the microbiota is described as being restored after tagatose treatment. Whilst the data does show increased alpha diversity in the treated group. The composition of microbiota is different from the untreated control group. Suggesting bacterial diversity has been restored but the composition of bacterial species has been altered.

Response 5: According to the reviewer’s suggestion, the sentence had been revised in the revised manuscript (page 10, line 315; page 11, line 352-356; page 13, line 454-456).

Point 6: The discussion is quite descriptive of the results in nature and would benefit by placing the results in context with published literature more. Especially, what is known about the use of carbohydrate supplementation during constipation and how this study adds to this.

Response 6: According to the reviewer’s suggestion, the relevant information had been added in the revised manuscript (page 9-10, line 263-271; page 10, line 293-296, line 300-308).

We look forward to hearing from you again.

With kind regards,

Hong Wang, Ph. D.

Professor

Guangdong Provincial Key Laboratory of Food Quality and Safety

College of Food Science

South China Agricultural University

Guangzhou 510642, China

Reviewer 2 Report

Dear authors,

Thank you for your efforts in this field.

Now it is common sense that the interaction between gut bacteria and host organs is very important to human health. In this concept, gut bacteria modulation is a significant strategy and application for human health improvement. Probiotics and prebiotics are major candidates for the bacterial therapeutics based on the gut bacteria modulation. In particular, carbohydrates are very effective materials as prebiotics. Therefore, I think this new approach using D-tagatose is very valuable.

However, I could not found appropriate and enough scientific evidence and approach for the author's critical conclusion that 'D-tagatose treatment is beneficial for restoring gut microbiota.'

I need to follow information to consider your conclusions:

For comment on the 'gut microbiota restore', please show me any difference of gut microbiota community between 'blank' and 'model'. Not enough only alpha and beta diversity analysis. What is dysbiosis status in your model? And then, what is the core bacteria?    You also can analyze the comparison of bacterial community between the time points days 14 and 28 for your model dysbiosis, but not shown. For comment on the 'constipation-relieving', please suggest a mathematical correlation between any biological data and gut microbiota composition.

Author Response

Dear Editor/Reviewers:

We are very grateful for your helpful suggestions for revision (Manuscript ID: ijms-617693). The manuscript has been carefully revised according to the editor and reviewers’ comments. All the changes are marked in red fonts throughout the revised manuscript. The issues raised by the editor and reviewers have been addressed as follows. Page, line or figure numbers refer to the revised manuscript. 

Point 1: For comment on the 'gut microbiota restore', please show me any difference of gut microbiota community between 'blank' and 'model'. Not enough only alpha and beta diversity analysis.

Response 1: In addition to the alpha and beta diversity analysis, the differences in the level of phylum, family and genus were also analyzed, and the specific contents had been added in the revised manuscript (page 5, line 156-168; page 5-6, line 181-194; page 6-7, line 211-219).

Point 2: What is dysbiosis status in your model? And then, what is the core bacteria?

Response 2: The dysbiosis status means that the abundance of the whole declines and difference of flora, mainly reflected by alpha diversity and beta diversity in the model group (Food & Function, 2019, 10(3): 1513-1528; Biomedicine & Pharmacotherapy, 2017, 96: 1075-1081). The core bacterium was regarded as the bacterium with the highest average relative abundance. At the phylum level, the core bacterium was Bacteroidetes, which were 77.97 %, 82.90 % and 69.31 % in the blank, model and tagatose group, respectively. At the family level, the core bacterium was BacteroidalesS24-7group, and its abundance in the blank, model and tagatose group were 45.81 %, 49.68% and 33.55 %, respectively. The core bacterium of the genus level was Bacteroides with the levels of 13.42%, 10.86% and 18.12% in the blank, model and tagatose group, respectively. The relevant information had been added in the revised manuscript (page 5, line 156-158, line 181-183; page 6-7, line 211-213).

Point 3: You also can analyze the comparison of bacterial community between the time points days 14 and 28 for your model dysbiosis, but not shown.

Response 3: Thank you for the review’s suggestion. In this study, referred to the methods of other studies(Food & Function, 2017, 8, (5), 1966-1978; Journal of Functional Foods 2017, 38, 486-496; Biomedicine & Pharmacotherapy, 2017, 96: 1075-1081), we mainly focused on investigating the effect of tagatose on the improvement of intestinal flora caused by constipation on the last day. Nevertheless, the suggestions of the reviewers were enlightening to us. In the following work, the comparison of bacterial community between the time points of day 14 and day 28 will be conducted.   

Point 4: For comment on the 'constipation-relieving', please suggest a mathematical correlation between any biological data and gut microbiota composition.

Response 4: According to the reviewer’s suggestion, the corresponding information had been added in the revised manuscript (page 8-9, line 239-261) (Figure 9).

We look forward to hearing from you again.

With kind regards,

Hong Wang, Ph. D.

Professor

Guangdong Provincial Key Laboratory of Food Quality and Safety

College of Food Science

South China Agricultural University

Guangzhou 510642, China

Round 2

Reviewer 2 Report

Thank you for the author's efforts.

I hope the article will improve our understanding of the scientific field.